# Spatio-Temporal Patterns of Fitness Behavior in Beijing Based on Social Media Data

**Bin Tian** [1,2] **, Bin Meng** [1,2,]*** , Juan Wang** [1,2]**, Guoqing Zhi** [1,2]**, Zhenyu Qi** [1,2]**, Siyu Chen** [1,2] **and Jian Liu** [3]

1   College of Applied Arts and Sciences, Beijing Union University, Beijing 100191, China;
    191070510112@buu.edu.cn (B.T.); wangjuan@buu.edu.cn (J.W.); 191070510111@buu.edu.cn (G.Z.);
    qizhenyu2013@gmail.com (Z.Q.); 20201070510119@buu.edu.cn (S.C.)
2   Laboratory of Urban Cultural Sensing & Computing, Beijing Union University, Beijing 100191, China
3   College of Resource Environment and Tourism, Capital Normal University, Beijing 100048, China;
    liujian@cnu.edu.cn
*   Correspondence: mengbin@buu.edu.cn

**Abstract:** Fitness is an important way to ensure the health of the population, and it is important to actively understand fitness behavior. Although social media Weibo data (the Chinese Tweeter) can provide multidimensional information in terms of objectivity and generalizability, there is still more latent potential to tap. Based on Sina Weibo social media data in the year 2017, this study was conducted to explore the spatial and temporal patterns of urban residents' different fitness behaviors and related influencing factors within the Fifth Ring Road of Beijing. FastAI, LDA, geodetector technology, and GIS spatial analysis methods were employed in this study. It was found that fitness behaviors in the study area could be categorized into four types. Residents can obtain better fitness experiences in sports venues. Different fitness types have different polycentric spatial distribution patterns. The residents' fitness frequency shows an obvious periodic distribution (weekly and 24 h). The spatial distribution of the fitness behavior of residents is mainly affected by factors, such as catering services, education and culture, companies, and public facilities. This research could help to promote the development of urban residents' fitness in Beijing.

**Keywords:** social media; fitness behavior; LDA model; geodetector; Beijing spatio-temporal patterns; Weibo

## 1. Introduction

Enhancing physical health is the top priority for stabilizing national development and improving people's quality of life. Physical health is largely influenced by individual biological factors (e.g., disease, heredity, gender, age) [1,2]. Fitness behavior is considered to be one of the most changeable influencing factors [3]. Moreover, in 2018, the World Health Organization concluded that non-communicable diseases (cardiovascular diseases, cancer, chronic respiratory diseases, and diabetes) are by far the leading cause of death worldwide, accounting for 71% of cases occurring globally. The four main risk factors driving the large increase in the number of diseases are smoking, harmful alcohol consumption, lack of exercise, and an unhealthy diet [4]. This shows that being physically active is a powerful way for residents to stay fit and healthy. Lack of physical activity is now a global pandemic [5,6], with about 31.1% of adults lacking sufficient physical activity [7]. Inactivity rises with age and is increased in high-income countries. In Europe, despite a 72% growth ratio in the number of fitness users over the past decade, almost 90% of European adults are still not users of fitness services [8]. In China, only 2.98% of the population participates in health clubs [9]. Therefore, active promotion of the development of fitness in the population is urgent, and we need to raise the attention placed on fitness behavior.

A number of scholars have conducted extensive research in the field of fitness behavior. Some scholars have taken a fitness population perspective, and more related studies have

focused on the differences in the fitness behaviors of different populations, such as the elderly, children, men, women, the rich, and the poor [10–14]. From the perspective of fitness venues, relevant research has paid greater attention to the transport range, accessibility, distribution characteristics, utilization rate, carrying capacity, and other influencing factors of these locations [15–20]. Some geographers are more concerned with the changes in fitness spaces within cities, and they usually considered elements, such as land use, neighborhood environment, accessibility, and population density, as the main factors influencing the changes in the fitness space [21–25]. These studies provided a solid foundation for revealing the phenomena and characteristics of fitness behavior and enriching the fitness life of residents. However, most of these studies were based on questionnaire or statistical data and were restricted by factors, such as the questionnaire design and statistical limitations, and affected by subjective factors, such as the recall and skills of the surveyors involved. As the temporal and spatial scale of the sample coverage was often small, there was a certain risk regarding the reliability of the resulting data and conclusions [26–29].

With increasing public participation, social media has become an important means through which residents can share their daily lives with one another. As a result, a large amount of social media data is generated. Social media data are rich in time-dimensional information, spatial-dimensional information, and textual information, and have great advantages for studying residents' behavior. Applying social media data in research can effectively make up for the number of samples and objectiveness that could not be achieved by previous data collection and survey methods. Many scholars have used social media data to focus on health-related topics, and have conducted extensive research exploring areas, such as air pollution, public health, natural disasters, environmental exposures, fitness spaces, and traffic safety [30–38]. For instance, using social media data, some studies have analyzed the spread of social media-based fitness guidance in the virtual space [39]. They combined social media data with urban point of interest (POI) data to study the spatial distribution and influencing factors of urban fitness spaces [40].

At present, there are few comprehensive studies of residents' fitness behavior using multidimensional information data. This study integrates time-dimensional information, spatial-dimensional information, and semantic information. Such integrated data can identify not only the user's spatial location but also the user's feelings through text analysis techniques [41], with the ability to observe the overall fitness status of users in the study area, summarize fitness characteristics, and explore influencing factors. Specifically, this research aimed to: (1) extract different fitness behaviors based on social media data using text processing techniques; (2) analyze the spatio-temporal patterns of the different fitness behaviors of residents based on temporal distribution features, spatial distribution features, and semantic features; and (3) explore the factors affecting the spatio-temporal patterns of different fitness behaviors.

## 2. Data and Methods

### 2.1. Study Area

This study considered the area within the Fifth Ring Road of Beijing as the research area. This area covers less than 5% of the total area of Beijing but has nearly half of the population of Beijing. At the end of 2017, the permanent population density in the Fifth Ring Road in Beijing was about 11,000 people per square kilometer, which was more than 8 times the average population density of the city.

### 2.2. Data and Pre-Processing

The social media data used in this study was obtained from the Sina Weibo platform. Sina Weibo is one of the most important social media platforms in China. According to the Weibo User Trends Report released in 2020 by Weibo Data Center (http://data.weibo.com/datacenter/recommendapp, accessed on 10 January 2021), as of September 2020, Weibo had 511 million monthly active users with 224 million daily active users. According to the research needs, this study has made targeted improvements to data acquisition methods.

Sina Weibo official API and web crawler tools were used to rawl Weibo data in Beijing. A total of 13,343,209 individual Weibo data items were obtained in the research area in 2017 (i.e., time, location, user UID, text information). A total of 1,973,596 Weibo users were involved.

To extract the fitness behavior of residents in Beijing, all of the data was first screened. Table 1 shows the preliminary screening rules and their processing purposes. According to the preliminary screening rules, more than 250,000 Weibo data points related to the fitness behavior of residents in the research area were obtained. Secondly, the text content of the Weibo data was cleaned. A custom dictionary for cleaning the data for fitness behavior was developed using a stop words database and a keyword thesaurus, and useless information, such as emojis, advertisements, and lotteries, in the Weibo text were removed. Based on this, we used the Bidirectional Encoder Representation from Transformers (BERT) deep learning algorithm to further extract fitness-related Weibo data. Finally, we removed the Weibo data with missing text information or other attribute information to obtain a total of 95,919 Weibo data points. A total of 31,700 Weibo users were involved.

**Table 1.** Preliminary screening rule chart.

| Preliminary Screening Rules | Objective |
| --- | --- |
| The target users sent less than 800 posts in 2017 | Excluding advertising account, marketing account and other non-personal account microblog |
| Users have posted in Beijing for three months or more | Ensure that social media users are residents of Beijing |
| Text content includes fitness related keywords (such as ball, cycling, running, fitness, etc.) | Exclude a large number of posts unrelated to fitness behavior |

Point of interest (POI) data from 2019 in the study area were also sourced from the Gaode Map platform. It included 12 main categories (restaurants, shopping, accommodation, science, education, culture, etc.). In this study, these data were mainly used to conduct an attribution analysis of fitness behaviors and to explore the influencing factors that affect different fitness behaviors.

The administrative division data used in this study were downloaded from National Geomatics Center of China (http://www.ngcc.cn/ngcc/, accessed on 12 March 2020).

*2.3. Research Methods*

2.3.1. Research Framework

This study constructed a framework for studying residents' fitness behavior based on natural language processing, spatio-temporal analysis, and statistical analysis (Figure 1). It is able to integrate time-dimensional information, spatial-dimensional information, and textual information to explore the spatio-temporal patterns of residents' fitness behaviors. Specifically, firstly, the Weibo data were pre-processed to obtain Weibo data related to fitness behaviors. Secondly, we classified the Weibo data by topic to obtain Weibo data with more fitness characteristics. Thirdly, we explored the spatio-temporal patterns and related laws of fitness behaviors from the three perspectives of time, space, and semantics. Finally, the influencing factors of the pattern were explored, and suggestions made to address them.

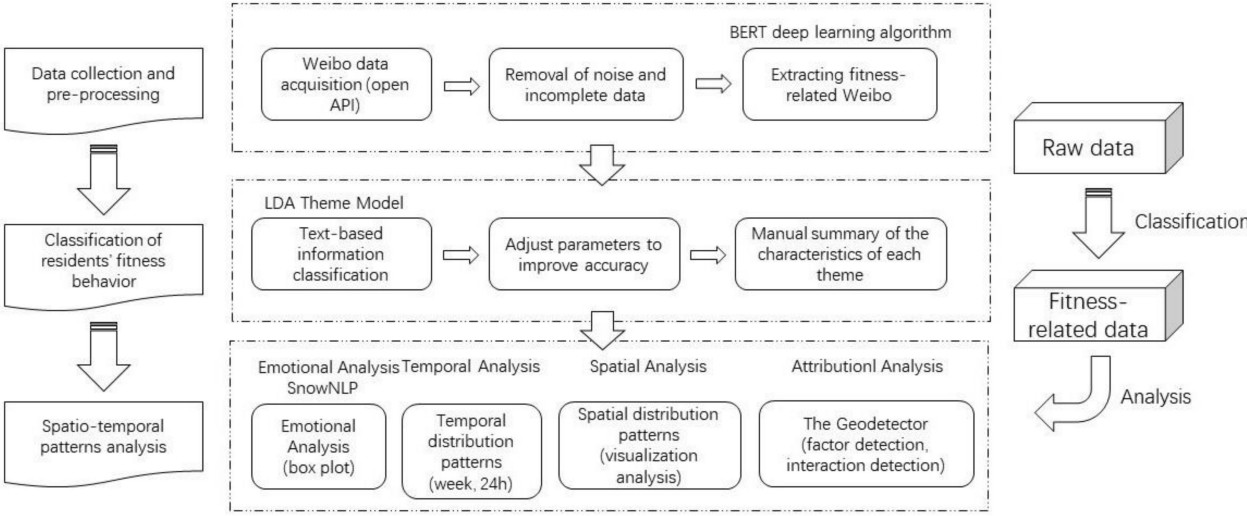

**Figure 1.** Framework of residents' fitness behavior research based on social media data.

### 2.3.2. Text Content Recognition

BERT (Bidirectional Encoder Representation from Transformers) is a language model released by Google in November 2018. It can greatly improve the accuracy of text recognition and is a major breakthrough in natural language processing (NLP) [40]. We used the BERT model, which complies with the python language, and combines FastAI technology [42], to realize the semantic processing of massive microblog text data and extract microblog data related to residents' fitness behavior. The specific implementation process included manual labeling of the training data, machine learning to build the training set, data iterative training to improve the accuracy, training of all microblogging data, and the storage of the results data. By importing a large number of manually labeled Weibo data and multiple iterations, the final recognition accuracy of this study was more than 93.8%.

### 2.3.3. Text Theme Detection

LDA (latent dirichlet allocation) is a well-known theme model, proposed by David M. Blei et al. [43], that has been widely applied in the processing of big data [44,45]. It is a typical unsupervised learning algorithm that does not require a large number of manual annotations of the training set during training. The core formula is shown in Equation (1):

$$P(w|d) = P(w|t) \times P(t|d) \tag{1}$$

where $P(w \mid d)$ refers to the probability that the word $w$ appears in the document $d$, $P(w \mid t)$ refers to the probability that the word $w$ may appear in the topic $t$, and $P(t \mid d)$ refers to the probability that the topic $t$ appears in the document probability in $d$.

The specific implementation process includes setting deactivated words and custom words based on text information, analyzing Weibo data, setting the appropriate number of categories based on the results, and storing the output results. This study used the python language to call the LDA model to apply subject classification to the Weibo text, and hence provide structured thematic data for the analysis of the characteristics of the fitness behavior of residents in Beijing.

### 2.3.4. Text Sentiment Extraction

Text sentiment analysis, also known as sentiment orientation analysis, refers to the process of analyzing, processing, summarizing, and reasoning texts with subjective sentiment orientation [46]. Generally, two methods of sentiment analysis exist: one is based on machine learning, which uses classification analysis to solve sentiment analysis, which requires a large high-precision training data set; and the other is based on sentiment dictionary, through the calculation of the sentiment score of each word and phrase is comprehensively

calculated for the sentiment value of the data, which requires a large number of labeled sentiment dictionaries. In this study, considering the characteristics of the data and the scalability of the research framework, a sentiment dictionary-based method was used to perform sentiment analysis of the Weibo data.

SnowNLP is a Chinese processing tool based on python language. This study used the sentiment analysis function of this tool to extract the sentiment value of the fitness microblog text information. The specific implementation process includes enriching the sentiment dictionary based on text information, establishing a dedicated sentiment dictionary of the fitness behavior, calculating the sentiment values of all Weibo data, and storing the calculation results. The result of the operation indicated the sentiment value of each Weibo, and used a number from 0 to 1 to indicate the probability of whether it was biased towards positive or negative sentiment (0 means negative, 1 means positive).

### 2.3.5. Geodetector

To further analyze the causes of the spatial characteristics of the fitness behavior of the residents in the research area, this study used geodetector tools for attribution analysis. Geodetector was developed by Wang Jinfeng and others. It consists of risk detectors, factor detectors, ecological detectors, and interaction detectors. It is a set of statistical methods that are used to explore geographic spatial differentiation and find its explanatory variables [47]. One of the advantages of geodetector over traditional statistics is that it can study the influence of the coupling between two factors on the dependent variable. The fitness behavior of residents is affected by factors, such as transportation and public facilities, and there are also coupling effects among the influencing factors. The use of geographic detectors effectively revealed the influencing mechanisms of the distributions of the fitness behavior of residents.

## 3. Results

### *3.1. Spatial-Temporal Patterns of Overall Fitness Behaviors*

To better reveal the spatial distribution of the fitness behaviors of residents in Beijing, this study analyzed the overall fitness Weibo data in terms of space and time. From the perspective of the spatial distribution (Figure 2), we constructed a grid of 1 km × 1 km within the 5th Ring Road in Beijing, and counted the number of fitness behaviors in each grid to more intuitively discover the overall distribution characteristics of residents' fitness behaviors. The fitness behaviors of residents in Beijing are generally distributed in multiple centers, with a wide coverage in the northern city and obvious clustering areas in the southern city. Specifically, areas with dense fitness activities are mainly concentrated in densely populated residential areas, such as Fengtai Town, Majiapu, Xiluoyuan, Songjiazhuang, Chaoyang District, Beitaipingzhuang, and Yuanda Road in Chaoyang District Regions and so on, which are labeled with red circles in Figure 2. Secondly, there are more distributions in large parks, such as the Olympic Forest Park. On the contrary, in some working areas where Beijing urban residents are relatively concentrated, such as CBD and Zhongguancun, the concentration is not high.

From the perspective of the time distribution (Figure 3), spring and summer are the seasons in which residents most frequently exercise. Over the year, residents exercise the most in March. From the perspective of working days, Wednesday is the day on which residents most frequently exercise. The number of fitness behaviors peaks twice in a day. The specific performance is Beijing residents generally start fitness activities at 5 a.m., and reach a small climax at 9 a.m. After that, the fitness behavior continues to increase, and reaches the peak of the day at around nine o'clock in the evening, and then it drops rapidly.

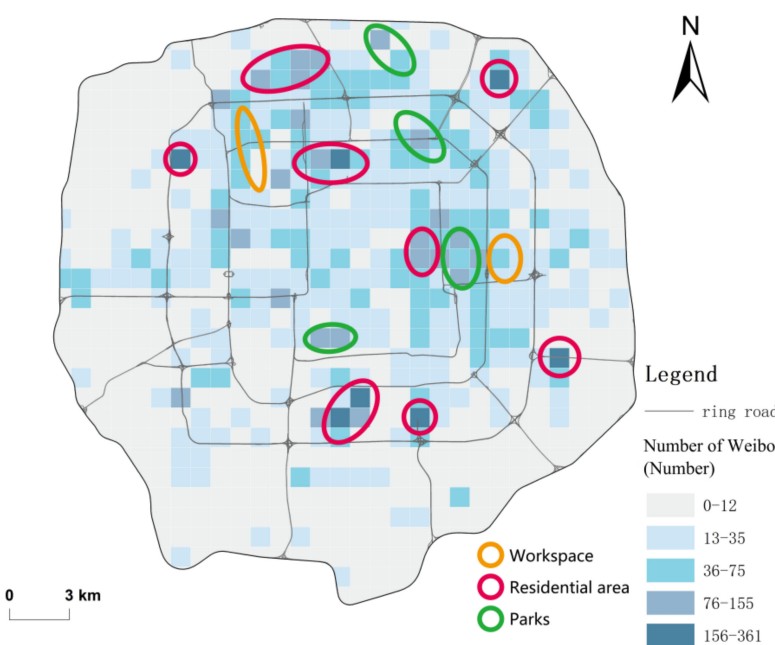

**Figure 2.** Spatial distribution of fitness behavior themes extracted form Weibo data.

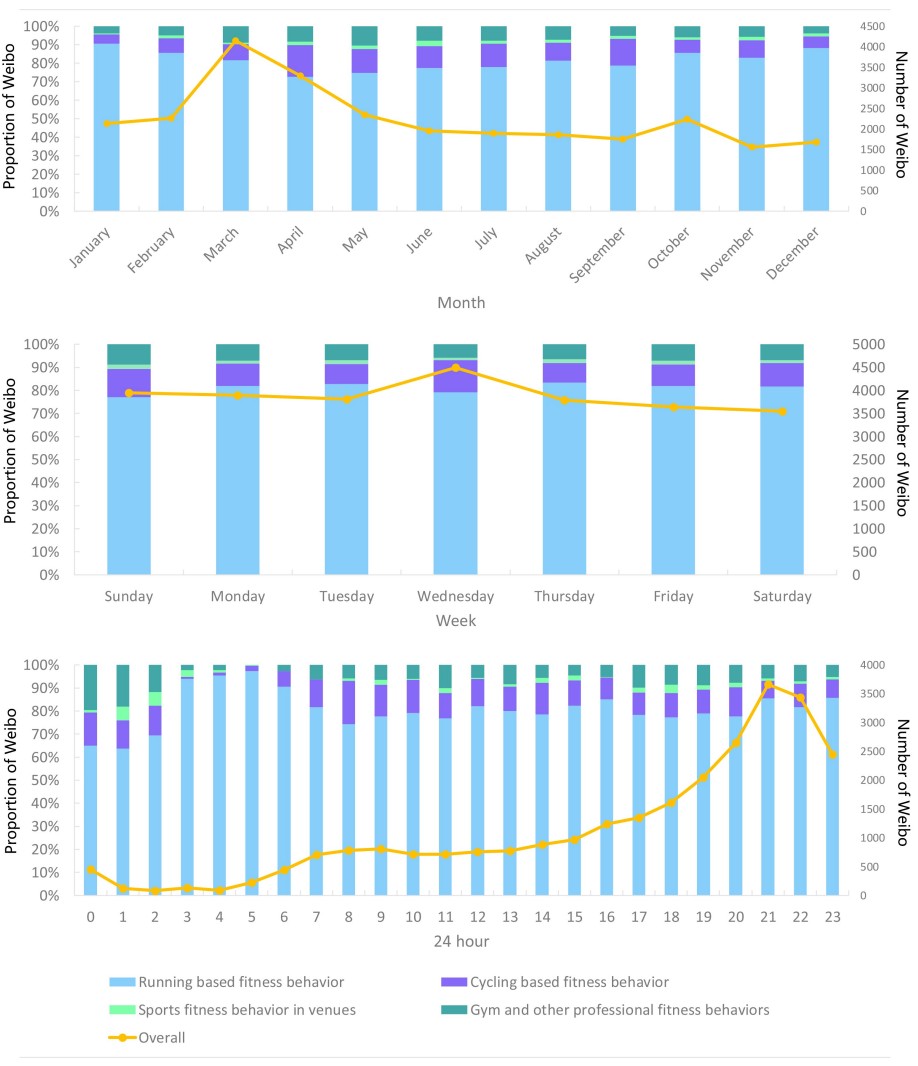

**Figure 3.** Temporal distribution of fitness behavior themes extracted form Weibo data.

### 3.2. Thematic Analysis

This study used an empirical setting method to determine the number of fitness behavior topics, including viewing the subject terms of the classification results, comparing whether the differences between different results are obvious, etc., after multiple tests and calculation of the optimal classification results. The microblog content related to the fitness behavior of Beijing residents was processed using LDA theme model technology. After repeated tests, it was finally determined that the best results were obtained by dividing it into four theme categories. Figure 4 shows the effect of the topic classification results on a two-dimensional plane after multiple dimensionality reductions. Among them, the size of each circle represents the number of samples contained in the different topics, and the differentiation between different topics is represented by the distance between different circles. The classification results show that when the number of topics is four, fitness behaviors can be divided into four categories, and there are obvious differences between the categories. Based on this, we obtained a total of 14,314 Weibo data with obvious thematic tendencies.

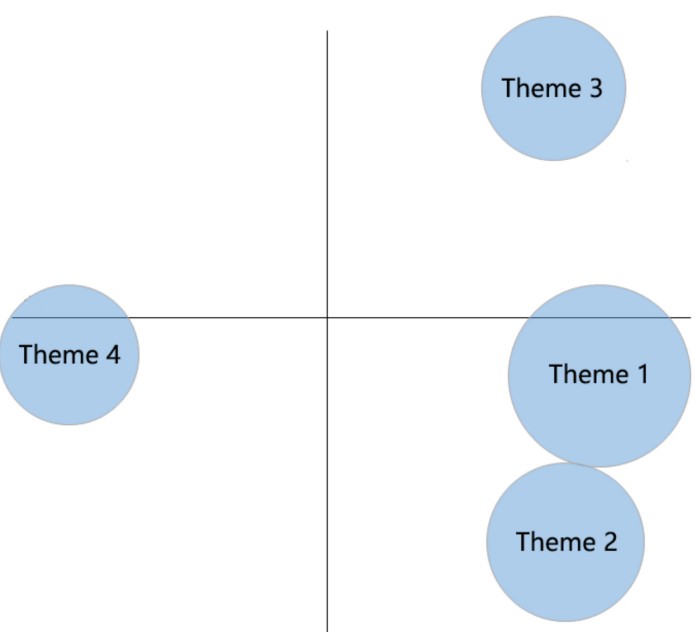

**Figure 4.** LDA Intertopic Distance Map (via multidimensional scaling).

In-depth analysis of the semantic characteristics of the high-frequency keywords of the four themes was used to condense the behavioral characteristics of each theme. The verb "run" appeared in the high-frequency keywords of theme 1, combined with encouraging words, such as "clock in" and "start", for which the characteristics are more obvious, so the behavioral characteristics were summarized as "running-based fitness behavior". The verb "cycling" appeared in the high-frequency keywords of the second theme, and fitness-related words, such as "kilometer" and "hold on". It can be clearly seen that this type is mainly based on long-distance cycling fitness, so the summary of its behavior was characterized as "cycling-based fitness behavior". The verb "play ball" appeared in the high-frequency keywords of topic three, and the qualifying words "today" and "minute". It can be inferred that such behaviors are performed in professional sports venues, such as basketball courts, table tennis halls, and badminton halls. Therefore, the behavioral characteristics were summarized as "sports fitness behavior in venues". The noun "gym" appeared in the high-frequency keywords of theme 4, and in words expressing emotions, such as "cheer", "oneself", and "feeling". It can be inferred that this type of behavior is performed in areas, such as the gym, using professional fitness equipment. The fitness behavior, therefore, summarized its behavioral characteristics, such as "gym and other

professional fitness behaviors". The number of the four themes in the total sample and the keywords are shown in Table 2.

**Table 2.** Keywords and types of LDA themes.

| Subject Category | Percentage of Total Sample Size | Keywords | Fitness Activity |
|---|---|---|---|
| Theme One | 32.5% | Exercise, clock in, run, start, hold on | Running-based fitness behavior |
| Theme Two | 26.8% | Fitness, cycling, clock in, kilometer, hold on, start | Cycling-based fitness behavior |
| Theme Three | 21.9% | Today, clock in, now, play ball, minutes | Sports fitness behavior in venues |
| Theme Four | 18.8% | Cheer, hour, gym, oneself, feeling | Gym and other professional fitness behaviors |

*3.3. Emotional Evaluation of Fitness Behavior*

The level of emotional value can directly reflect the feelings of residents when they perform fitness behaviors, and it is also an important index used to enrich the study of the characteristics of the spatio-temporal distribution of residents' fitness behaviors. In response to this analysis, this study used manual screening and keyword extraction to eliminate the check-in Weibo automatically generated by APP from all Weibo data. Its purpose was to reduce the impact of such Weibo data on the emotional analysis of residents' fitness behaviors.

In this study, sentiment analysis tools were used to calculate the sentiment value of each Weibo, and the sentiment value was used to indicate the mood of residents during the fitness behavior. According to the results of the subject classification, we constructed box plots of the sentiment values for each category of fitness behavior. In the plots, here the line in the middle represents the median, and the lines toward the sides represent the quartiles and the edges, respectively. Points outside the edges are outliers. The sentiment value in Figure 4 is a probability function, where the closer it is to 1, the more positive the sentiment tends to be, and the closer it is to 0, the more negative the sentiment tends to be. The use of this method can intuitively reflect the emotional characteristics of residents regarding the four fitness behaviors. The results show that residents have better overall fitness experiences (Figure 5). Among them, performing physical fitness behaviors in venues generally results in a better fitness experience and a more comfortable mood. In contrast, sometimes residents do not have a good fitness experience when using professional fitness equipment in the gym or at home.

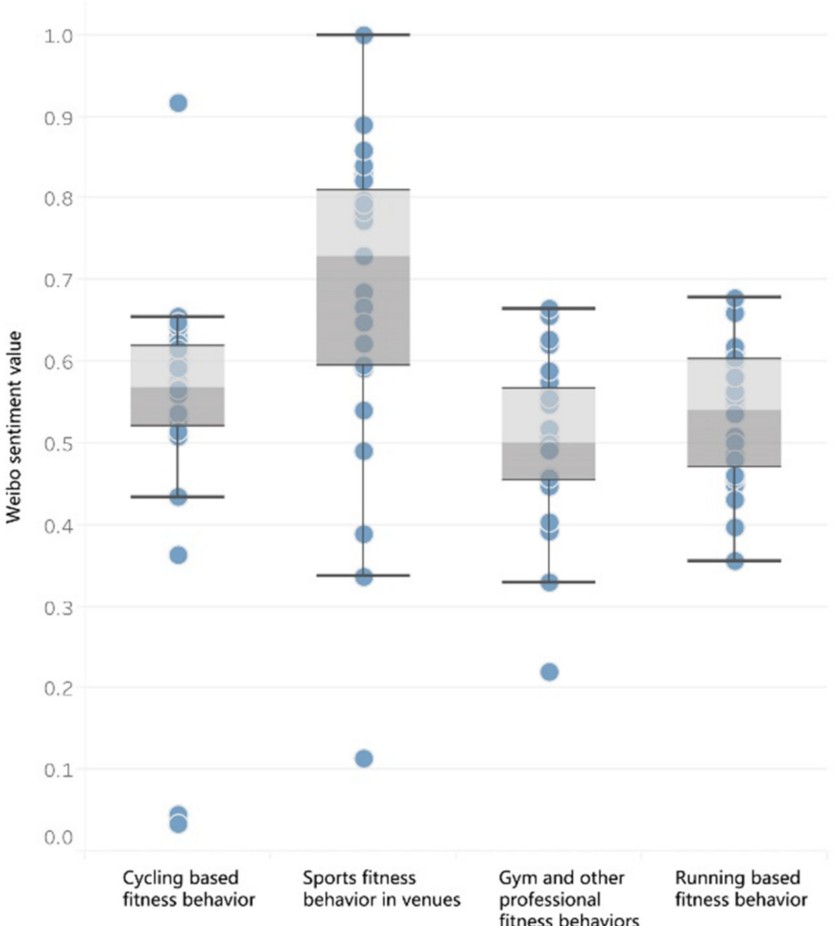

**Figure 5.** Emotional mean value of the four thematic behaviors.

*3.4. Spatio-Temporal Patterns of Each Fitness Behavior*

Based on the above research results, this study also separately interpreted the microblog text information of the four fitness behaviors and produced the corresponding nuclear density distribution map (Figure 6). This is a visual analysis of the results, which allows an intuitive visualization of the results to decipher their intrinsic properties and spatial distribution characteristics.

3.4.1. Theme 1: Running-Based Fitness Behavior

The main representative of theme one is running-oriented fitness behaviors. Most of these behaviors are residents expressing their feelings after long-distance running, or recording their every exercise by clocking in. In this part of the check-in data, about half of the check-in records were automatically generated using sports apps. It can be seen that sports apps have a greater impact on residents in this type of fitness behavior. A small number of fitness behaviors involve other sports related to running at the same time, such as skipping rope and hiking.

From the perspective of the spatial distribution, the overall fitness behavior of this theme presents a distribution pattern of "more gathering areas, more east and less west". This type of fitness behavior is mainly concentrated near residential areas, such as near Songjiazhuang and Wangjing. Secondly, parks near residential areas are mostly distributed, such as Ritan Park. In addition, it is also distributed near some colleges and famous attractions.

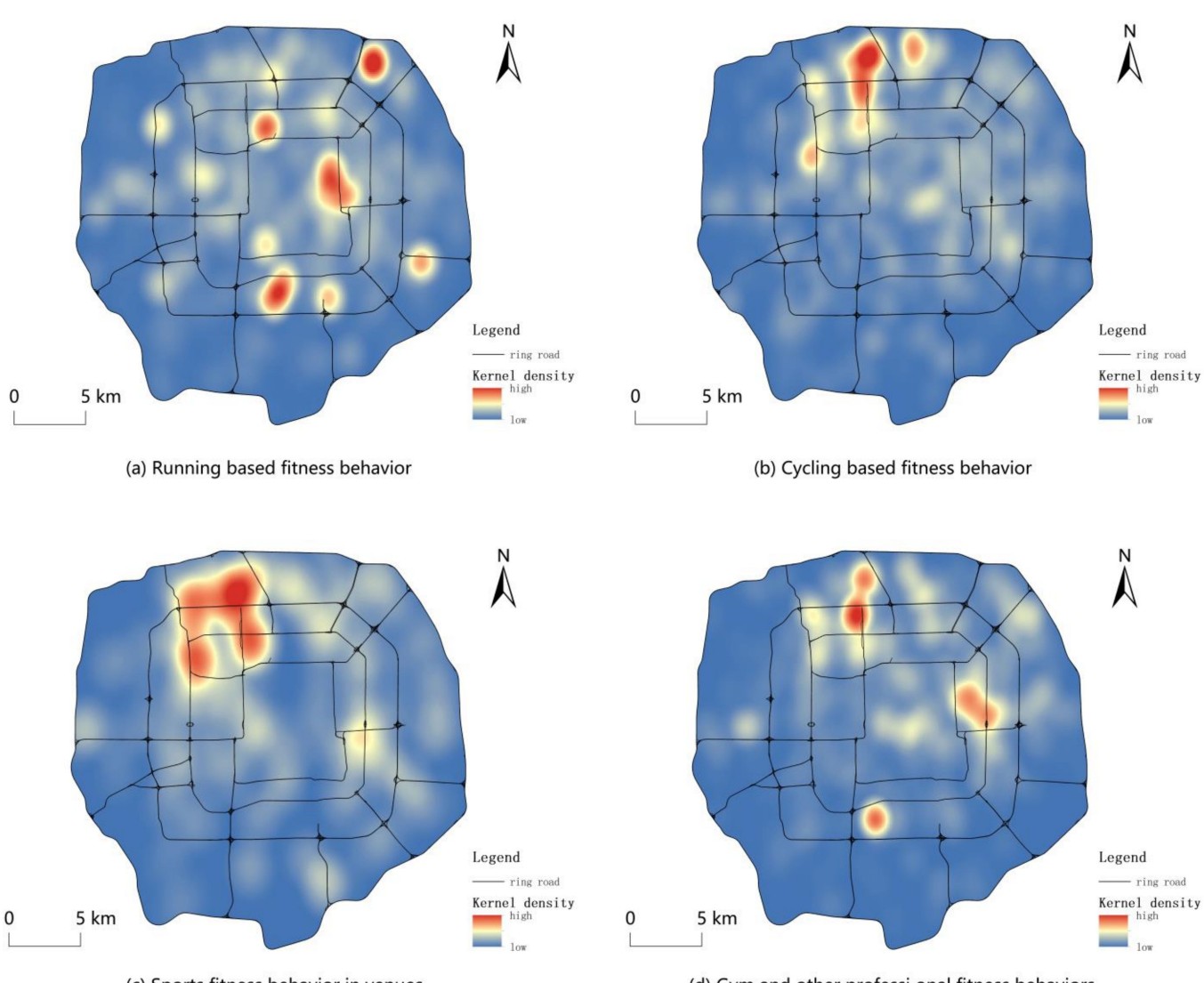

**Figure 6.** Nuclear density map of fitness behaviors for each theme.

### 3.4.2. Theme 2: Cycling-Based Fitness Behavior

Theme 2 mainly represents fitness behaviors based on cycling. The main types include urban cycling, cycling to scenic spots, night cycling fitness, and the popular shared bicycle cycling check-in. Similarly, there are many check-in records automatically generated by residents using sports apps after exercising. In addition, some fitness-related topics initiated in Weibo are also some of the important factors affecting residents' fitness behaviors.

From the perspective of the spatial distribution, the overall situation is "one center, two sub-centers, more in the north and less in the south". This theme fitness behavior is distributed in clusters around universities and parks, such as Capital Normal University, China Agricultural University, Beijing Language and Culture University, Beijing University of Aeronautics and Astronautics, Yuandadu Ruins Park, Olympic Forest Park, etc, and is mainly distributed between the North Fourth Ring to the North Fifth Ring.

### 3.4.3. Theme 3: Sports Fitness Behavior in Venues

Theme three mainly includes recreational fitness behaviors, such as basketball, swimming, and badminton. This type of fitness behavior has more stringent requirements on the venue than running and cycling. Sports, such as table tennis and tennis, need to be performed in professional fitness venues. In the relevant texts, many residents mentioned that they had to commute long distances to play ball. Therefore, some residents would

also perform other sports, such as cycling and running, while performing fitness behaviors related to this subject.

From the perspective of the spatial distribution, the overall appearance of the spatial clustering spreads from the center to the surroundings. The central area is located between Minzu University of China, Peking University, Liudaokou, and Beijing Normal University. The distribution feature of this theme is similar to that of theme 2, but it is concentrated more in the vicinity of universities. At the same time, there are also small gathering areas near the International Trade Center.

### 3.4.4. Theme 4: Gym and Other Professional Fitness Behaviors

Theme 4 mainly includes fitness behaviors that use professional equipment to train in the gym. The fitness content is richer than the first three themes. In addition to the emotional expression of residents after fitness, it also includes numerical records and fitness opinions using ellipsometers, dumbbells, and other equipment. Residents who do this kind of exercise tend to have more stringent requirements regarding their own health and body shape. Most of them have clear fitness programs and fitness intensity, and summarize details about any changes in their health and weight.

From the perspective of the spatial distribution, this type of fitness behavior mainly has three concentrated areas. This type of agglomeration area has a more significant feature, that is, it is mainly located near large residential areas and commercial centers, such as Liudaokou in the north, Guomao in the east, and Jiaomen West in the south.

### 3.4.5. Comprehensive Analysis

To analyze the four fitness behaviors together, we created a "hot-spot" map (Figure 7). It is a visual analysis result based on location information, which can discover obvious hotspot gathering areas. By observing the "hot-spot" map of the fitness behaviors of residents in the research area, we can clearly see that there are obvious hotspots of the different themes of fitness behavior and the distribution of hotspots varies greatly. (1) Overall, there was more activity in the north of the research area than in the south, with some significant hotspots in the area of Xueyuan Road-Lincui Road-Osen Park, near the International Trade Center and the Beijing Central Business District, and near large residential areas in the south and east. (2) Different fitness behaviors were unevenly distributed in space. Theme 1 showed discrete hot spots in the east and south of the area, and the fitness behaviors of theme 4 showed discrete hotspots in the center.

In summary, the aggregation of the centers of fitness behaviors in the research area was relatively high, and there was an obvious trend of activities gradually decreasing from the center of the aggregation to the surroundings. Additionally, a small number of high-density areas were also formed in some peripheral areas, and the overall distribution pattern was of large aggregation areas as the main body with scattered small aggregation areas.

We observed temporal pattern differences between the 4 fitness behavior themes using 2 time frames: 24 h and 1 week (see Figure 8). We normalized the number of Weibo for different time periods. The number of fitness behaviors is indicated in the graph by the size and color shades.

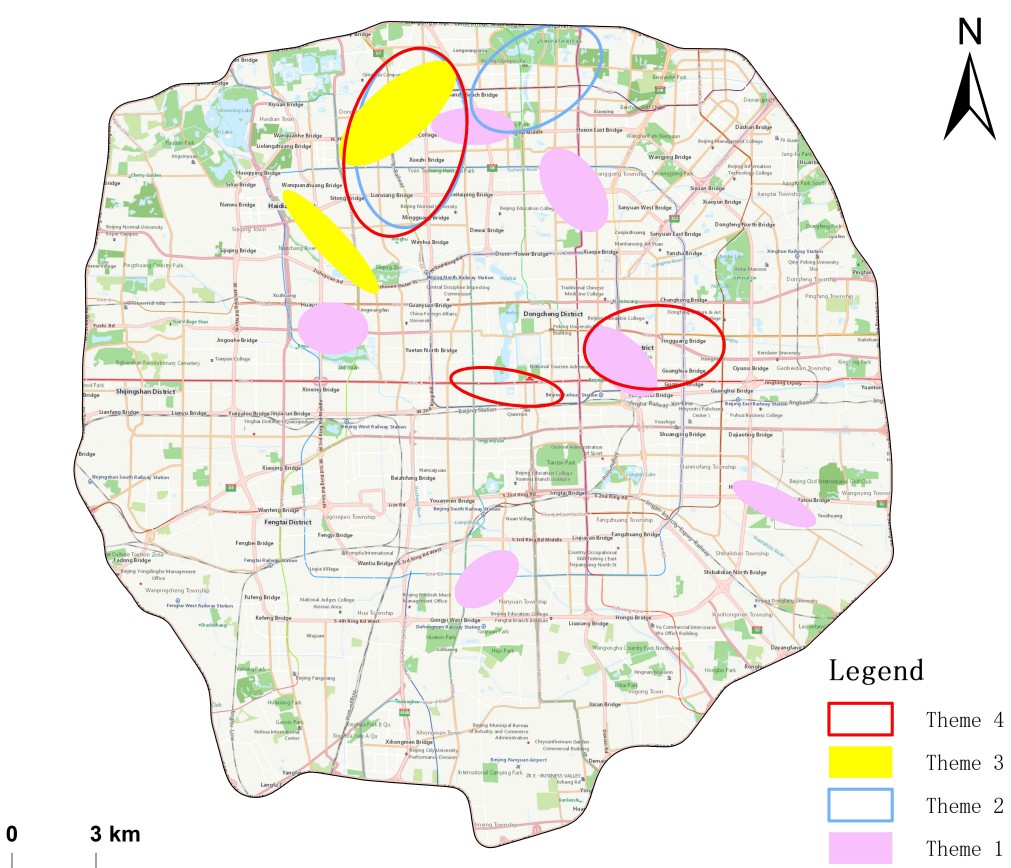

**Figure 7.** "Hot-spot" map of the four fitness behavior themes.

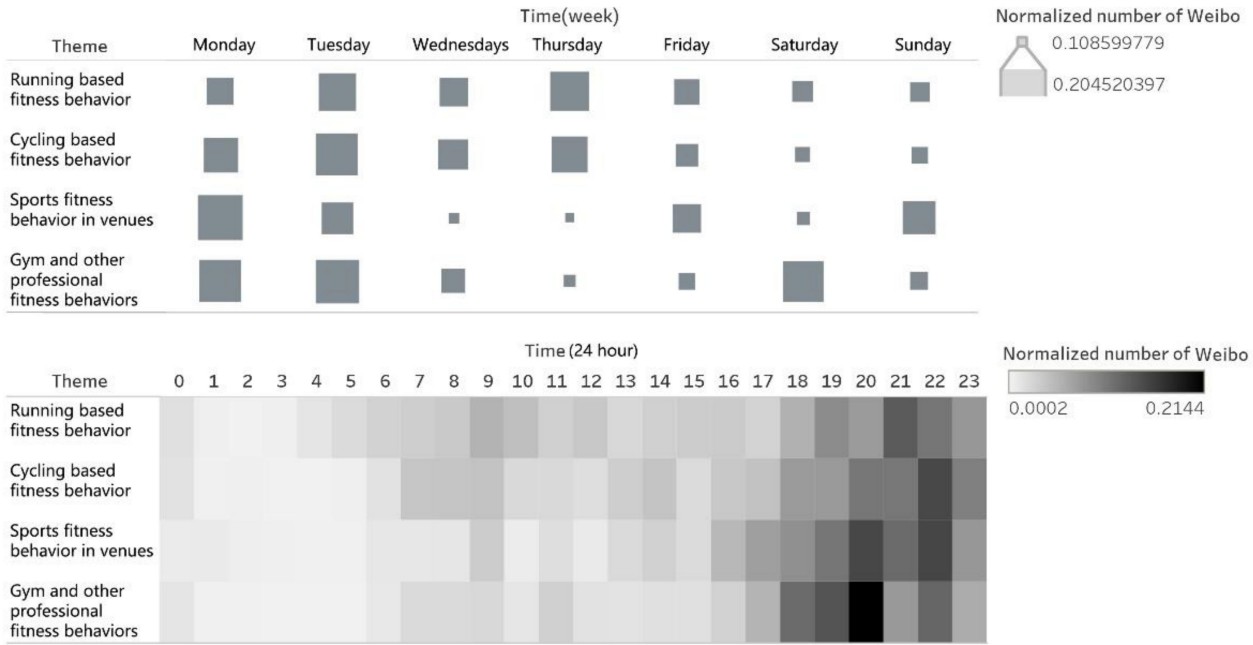

**Figure 8.** Temporal pattern of the four fitness behavior themes.

In relation to the 24-h daily cycle, residents' exercise time was mainly concentrated in the evening, reflected by the gradual increase in the number of exercisers from 17:00 in the afternoon, and reaching a peak at around 20:00 in the evening. Additionally, some residents also performed fitness behavior from 7 a.m. to 10 a.m. The types of fitness activities at this time were mainly theme 1 (running) and theme 2 (cycling).

Taking a week as the cycle, residents' fitness activity was mainly concentrated on Sunday and Monday. For the fitness behaviors of theme 1 and theme 2, which are less affected by the need for specific venues, their time distribution characteristics were similar, and were focused on Monday and Wednesday, although they had a high degree of participation on almost every day. For fitness activities that have specific needs regarding venues or facilities, there was a large difference in daily participation. The fitness behavior of theme 3 mainly took place on rest days, with partial distributions on Mondays and Thursdays. The fitness behavior of theme 4 mainly occurred on Sunday, Monday, and Friday.

### 3.5. Related Factors to Fitness Behavior

Based on the findings described above, this study used a geodetector tool to further explore relevant factors that affect the spatial differentiation of the fitness behavior of residents. First, an evaluation index system of the influencing factors was constructed based on a total of 14 explanatory variables in 6 categories, including various elements of the city, environmental conditions, land prices, traffic convenience, population distribution, and location conditions (see Table 3). Based on this, the factor detection and interaction detection functions of the geographic detector were used to reveal the influences on the spatial selection of the fitness behavior of residents in Beijing.

**Table 3.** Index system of influencing factors of the spatiotemporal differentiation of residents' fitness behavior.

| Influencing Factors | Explanatory Factors |
| --- | --- |
| Various urban elements | Number of food service facilities (X1), number of sports and leisure facilities (X2), number of public facilities (X3), number of business service facilities (X4), number of accommodation facilities (X5), number of educational and cultural facilities (X6), number of companies and enterprises (X7), number of wholesale and retail facilities (X8), number of residential service facilities (X9) |
| Environmental conditions | Distance to the nearest park (X10) |
| Land Price | Land value (X11) |
| Convenience of transportation | Distance to bus and subway stations (X12) |
| Population Distribution | Population density (X13) |
| Location conditions | Distance to CBD (X14) |

Use of the factor detection function of the geodetector can effectively explore the degree of influence of each influencing factor on the spatial choice of residents' fitness behavior (Table 4). The research results show that number of food service facilities, number of residential service facilities, and number of educational and cultural facilities are the main influencing factors regarding fitness behavior. At the same time, for further research on the fitness behaviors of different themes, we found that the influencing factors of the fitness behaviors of various categories are significantly different. The running-based fitness behavior is greatly affected by the distribution of residents' services, catering, and accommodation facilities, indicating that it is more dependent on supporting service facilities in nearby communities. The cycling-based fitness behavior is greatly affected by land price, education, and catering facilities. The sports fitness behavior in venues and the gym and other professional fitness behaviors are mainly affected by the distribution of educational and cultural facilities, but the former is also affected by the distribution of public facilities while the latter has greater requirements regarding the distribution of residential service facilities.

The interaction detection function of the geodetector can indicate whether the combined effect of two different factors will enhance or weaken the factor explanatory power for the dependent variable, and it can effectively reveal the impact of the two factors on the spatial choice of the fitness behavior of residents. The results suggest that (1) in this study, the explanatory power of the 14 influencing factors after pairwise interaction is greater than the explanatory power individually, which indicates that the fitness behavior of residents is

jointly restricted by the influencing factors of various dimensions, and any two influencing factors will enhance the factor explanatory power for the dependent variable; (2) overall, residents are influenced by whether there are catering service facilities available to them, such as restaurants and street snacks, when they exercise. Such areas are often accompanied by extremely high traffic flow or by densely populated areas; (3) in total, 12 indicators of 4 fitness behaviors were assessed using pairwise interactive detection (see Table 4). The results confirm that the different fitness behavior themes are influenced by different factors. For instance, the fitness behavior of theme 1 was more likely to be affected by two factors: the distribution of catering facilities and the population density, while the fitness behavior of theme 2 was more susceptible to the interaction of land prices and the distribution of educational and cultural facilities.

**Table 4.** Results of the attribution analysis.

| | Type Characteristics | Main Impact Factors | Secondary Impact Factors |
|---|---|---|---|
| **Factor Action Detection** | Overall | X1, X9, X6 | X4, X5, X11 |
| | Running based fitness behavior | X9, X1, X5 | X8, X6, X3 |
| | Cycling based fitness behavior | X11, X6, X1 | X12, X4, X2 |
| | Sports fitness behavior in venues | X6, X11, X1 | X4, X3, X12 |
| | Gym and other professional fitness behaviors | X6, X1, X11 | X4, X9, X12 |
| | **Type Characteristics** | **Interaction Factors** | **Factor Explanatory Power** |
| **Interaction Detection** | Overall | X1∩X3 | 0.446 |
| | Running based fitness behavior | X1∩X13 | 0.423 |
| | Cycling based fitness behavior | X6∩X11 | 0.438 |
| | Sports fitness behavior in venues | X5∩X6 | 0.354 |
| | Gym and other professional fitness behaviors | X3∩X6 | 0.432 |

## 4. Discussion

Based on social media data, this study integrated big data mining methods and spatio-temporal analysis methods to conduct relevant research on the fitness behavior of Beijing residents. In previous fitness-related studies, traditional data, such as questionnaires and interview data, were mainly used [21,22]. Most studies are limited by the subjectivity of traditional data, the amount of data, and other factors, and there are some risks regarding the reliability of the findings [26–29]. The social media data posted by users on social media platforms contains rich information on the dynamics of residents' behaviors, which is important for understanding the spatio-temporal patterns of residents activities in current society [48]. The application of social media data in research can effectively compensate for the sample size and objectivity, which is not possible with traditional data. Based on this, this study more objectively and reliably explored the spatio-temporal patterns of different fitness behaviors and related influencing factors of urban residents within the Fifth Ring Road of Beijing.

Firstly, it is necessary to study residents' fitness behavior from the perspective of time, space, and text. It has been shown that the analysis of fitness data, especially big data, is very important [49]. Text information mining methods using big data can efficiently process text information in social media data, and geographical spatio-temporal analysis methods can effectively analyze the spatio-temporal information contained within social media data. The combination of the two can be used to fully explore residents' fitness behavior patterns and characteristics hidden within massive data. This study constructed a complete and efficient framework for the extraction and analysis of fitness-related information from the Weibo social media platform.

Second, overall, the spatial distribution of fitness behavior among Beijing residents is uneven, with multiple gathering centers. Related studies have shown that environmental factors directly influence the distribution of fitness behaviors [21,50], which is similar to

the findings of this paper. Most of the fitness locations for Beijing residents are located near residential areas, schools, and other areas with abundant fitness facilities. Based on this, this study further analyzed the fitness behavior of different topics using the LDA topic modeling technique. The results of the sentiment analysis show that Beijing residents generally have better fitness experiences, but residents have poor fitness experiences in the gym. Therefore, improving the service facilities in such areas can effectively improve residents' fitness experiences. The results of the spatial analysis show that the gathering areas of fitness behaviors of different themes are mainly concentrated in the vicinity of universities and residential areas in the North Fourth Ring Road, and show the characteristics of being distributed in the north and less in the south. Indeed, there are differences in the spatial distribution of the fitness behaviors of the different themes as the characteristics of each type of fitness are different, and are affected by factors, such as time cost, economic conditions, place environment, and personal subjective wishes.

Third, from a temporal perspective, the fitness behavior of Beijing residents shows obvious cyclical characteristics. It has been found that the temporal distribution of fitness behaviors of Beijing residents is significantly seasonal, mainly concentrated in summer and distributed least in winter [41]. On this basis, this study also found significant differences in the weekly activity intensity and 24-h activity intensity of fitness behavior in the 4 different fitness activity themes, and they showed obvious temporal characteristics and periodic characteristics. Among them, the fitness behavior of theme 1 (running) and theme 2 (cycling) had a relatively even distribution regarding the weekly activity intensity and generally occurred during the day. The fitness behavior of theme 3 (in venues) and theme 4 (professional fitness) had relatively discrete distributions regarding the weekly activity intensity, and were mainly concentrated at night. These characteristics were mainly related to the type of fitness and its convenience. Fitness behaviors, such as running and cycling, are less affected by the venue and the environment, and the cost is low. Hence, they are very popular among residents. Additionally, the four fitness behaviors have a higher intensity of daytime activities on Monday (the first day of the working day), and the underlying influence mechanism is worth exploring.

Fourth, the spatio-temporal distribution of the fitness behavior of residents in Beijing is affected by multiple factors. Related studies have shown that the distribution of residents' fitness behavior is influenced by various factors, such as the neighborhood environment [22], place of residence [23], and built environment [24]. However, unlike previous single-factor studies, this study integrated multiple factors and used a Geodetector tool to detect the main factors influencing the fitness behavior of Beijing residents. The results of single-factor detection and interaction detection show that the spatial distribution of catering service facilities and public facilities generally had a high explanatory power for the fitness behavior choices of residents. Additionally, it was also affected by factors, such as land price, population density, and the number of companies. From this, we can see that when residents choose fitness locations, they will often choose fitness venues near restaurants. Additionally, the neighborhoods of bustling business districts is also a favorite fitness place for residents. However, from the perspective of the four types of fitness behaviors, the interaction among factors, such as accommodation facilities, educational and cultural facilities, land prices, and population density, also has a strong explanatory power. Therefore, rationally strengthening the construction of Beijing's urban fitness service facilities, especially in densely populated areas, and increasing the diversity of fitness service facilities near some residential areas in the Fifth Ring Road can effectively implement a reasonable distribution of Beijing's urban fitness service facilities, improve residents' fitness environments, and improve residents' quality of life.

This study conducted an observational study of resident fitness behavior using social media data. However, the study has some limitations. First, since the majority of Weibo users are young, there is bias in the Weibo data. Caution must be exercised when extending the results to the general population. Future research needs to combine social media data with traditional survey data to fill data gaps and improve the reliability of the data.

Secondly, due to the limitations of web crawling technology and the falsifiability of social account information, conventional methods cannot directly obtain accurate social attribute information, such as the age or gender of microblog users. Enriching the information dimension of participant attributes from a data mining perspective is suggested as a follow-up study. As much as possible, we can mine information on social attributes, such as the user's place of residence, place of work, age, and gender, or use deep learning algorithms to identify behavioral categories for each piece of social media data to provide reliable data support for further research. Finally, the present work is an observational study with no comparisons, correlations, or other statistical analyses regarding the fitness behavior of the recruited subjects. Follow-up studies can use potentially advanced methods from behavioral science and the rational use of some selection models. On this basis, we actively explored the characteristics of residents' behaviors, such as fitness, tourism, and travel, with a view to exploring more rational explanations for residents' behaviors.

## 5. Conclusions

This study constructed and implemented a complete spatio-temporal analysis framework for fitness behavior based on social media data, using a combination of big data mining techniques, such as FastAI and LDA, and geographic analysis methods, such as Geodetector and spatio-temporal analysis. This study found that, first, the overall fitness behavior of Beijing residents is unevenly distributed spatially, with a greater distribution near residential areas and schools. Second, the fitness behaviors of Beijing residents are grouped into four themes, namely "running-based fitness behavior", "cycling-based fitness behavior", "sports fitness behavior in venues", and "gym and other professional fitness behaviors". The first fitness theme is the favorite fitness category of the residents while the fourth fitness theme has the lowest number of residents. Third, when carrying out the fitness behaviors of the different themes, residents have different emotional evaluations. Residents have better fitness experiences in sports venues. Fourth, the different thematic fitness behaviors are have different hotspot areas regarding the spatial distribution of, and they are mainly concentrated in the northern part of the study area. Fifth, there is significant periodicity in the temporal distribution of the different fitness behavior themes (24 h and 1 week). Finally, the spatial distribution of the fitness behavior of residents is mainly affected by factors, such as catering services, education and culture, companies, and public facilities. The main influences on fitness behaviors vary by theme, for example, "running-based fitness behaviors" are strongly influenced by the interaction between catering services and the population distribution.

**Author Contributions:** Conceptualization, B.M., J.W.; Data curation, G.Z., Z.Q., S.C. and J.L.; Writing—original draft, B.T. All authors have read and agreed to the published version of the manuscript.

**Funding:** This research was funded by the National Natural Science Foundation of China (Grant Nos. 41671165, 51878052) and the Academic Research Projects of Beijing Union University (Grant Nos. ZK40202001, RB202101).

**Institutional Review Board Statement:** Not applicable.

**Informed Consent Statement:** Not applicable.

**Data Availability Statement:** Not applicable.

**Acknowledgments:** Thanks for the funding of "innovation project for students of advanced and sophisticated disciplines of Beijing Studies". Thanks to the anonymous reviewers and editor for their meticulous and patient review, their comments and suggestions were extremely helpful to the article.

**Conflicts of Interest:** The authors declare no conflict of interest.

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
