# Peer review of "Spatio-Temporal Patterns of Fitness Behavior in Beijing Based on Social Media Data"

_sustainability, doi:10.3390/su14074106_

Round 1

Reviewer 1 Report

This paper employs FastAI, Latent Dirichlet Allocation (LDA) and other text mining techniques coupled with GIS spatial analysis methods to study temporal and spatial patterns of fitness behavior of residents in Beijing, China, from the perspective of residents’ daily behavior using social media data.

Overall, this is a very nice and comprehensive study in terms of data, methodology and analysis. I enjoy reading the article and recommend it to be accepted with minor revisions. I merely have minor suggestions as follows:

1, the source of used data should be more explicitly given. More details about the sample size for analysis should be provided

2, Relevant literature has a lot of Chinese references that cannot be accessed. It is suggested to cite relevant English literature to highlight your contributions, which at least is available for other scholars

3, some typos exist in the manuscript, such as table 4, which should be from journal template. The authors should carefully check the writing again.

4, Besides the methods used in the paper, it is also interesting to further apply some choice modeling for the research topic to explore more behavioral interpretation besides only presenting the observed behavior. In this sense, some discussion and literature about potential advanced methods in behavioral science should be added for pointing out future study directions.

Prospect theoretic contributions in understanding traveler behaviour: a review and some comments

Revealing psychological inertia in mode shift behavior and its quantitative influences on commuting trips

Future transportation: Sustainability, complexity and individualization of choices

Cumulative prospect theory coupled with multi-attribute decision making for modeling travel behavior

Author Response

Thank you for your comments. Please see the attachment for specific responses.

Reviewer 2 Report

This is an interesting study  that used FastAI, Latent Dirichlet Allocation (LDA) and other text mining techniques coupled with GIS spatial analysis methods to study temporal and 19 spatial patterns of fitness behavior of residents in Beijing, China. The article is well written, and the results are presented in clear pattern. 

However, there are some areas where authors can think to improve. For example, the location of the study is Beijing, but within 5th ring road. However, it would be difficult for international audience to learn about Beijing. Thus, a short description of Beijing, "probably study background, or study location" would help the readers to understand the context better. 

Author Response

(The authors gave the same response as above.)

Reviewer 3 Report

General comments

The authors have clearly stated that the purpose of the study was to examine the temporal distribution of fitness behavior, the spatial distribution of fitness behavior and the main factors affecting fitness behavior in Beijing based on social media data. The paper is well-written and easy to follow. In my opinion, it adds considerable value to the promotion of an active lifestyle for the masses through regular physical activity and exercise in a free-living environment. Although the local focus of the present study, this work can enhance future attempts in similar research area conducted in a different as well as larger geographical scale. However, I have highlighted a few suggestions and concerns in my specific comments section (below) that need to be addressed aiming to improve this work. In summary, both introduction and discussion section need to be supported by additional evidence in order to present a robust article that provides readers with valuable remarks.

Specific comments

ABSTRACT

- Sina Weibo should be added to methodology as the selected social media platform.

- The following statement should be added in some way to the methodology: “This study takes the city within the Fifth Ring Road of Beijing as the research area. This area represents less than 5% of the total area of Beijing, but has nearly half of the 89 population of Beijing.”.

- Line 23: Substitute “get” with “obtain”.

  1. INTRODUCTION

- Nice work from the authors. I suggest adding a few sentences in the introduction aiming to connect the topic with the current status of the health and fitness industry in Asia, Europe, and worldwide. In addition, more information about the health state nationwide is needed (i.e., prevalence of inactivity, overweight/obesity, diabetes, hypertension, and smoking). To do that, please consider citing the following resources.

References:

  • Kercher VM, Kercher K, Bennion T, Levy P, Alexander C, Amaral PC, et al. 2022 Fitness Trends from Around the Globe. ACSMs Health Fit J 2022; 26(1): 21-37.
  • Kercher VM, Kercher K, Bennion T, Yates BA, Feito Y, Alexander C, et al. Fitness Trends from around the globe. ACSM’s Health & Fitness Journal 2021; 25(1), 20–31.
  • Batrakoulis A. European Fitness Trends for 2020. ACSMs Health Fit J 2019; 23(6): 28–35.
  • World Health Organization Noncommunicable Diseases Country Profiles 2018; World Health Organization: Geneva, Switzerland, 2018.
  • International Health, Racquet and SportsClub Association. (2019). The IHRSA Greater China Health Club Report. Boston (MA): IHRSA Publications.

https://www.ihrsa.org/publications/the-ihrsa-greater-china-health-club-report/

  1. RESULTS

- Line 175: Add a space before the parenthesis “(Figure 2)”. Check similar issues throughout.

- All Tables and Figures really support the readability of the paper. However, the inclusion of more specific information regarding the participants’ characteristics, especially age and gender, would be a very important improvement. If you have access to those data, please add them where needed.

- In some tables and figures, an empty line is needed before continuing with the text. Please check throughout.

  1. DISCUSSION

- Connect introduction with discussion through a few sentences presenting the current status of the health and fitness industry as well as the country’s health profile as suggested in a previous comment (see introduction’s comments and suggested references).

- Line 432: Substitute “shortcomings” with “limitations”. In this paragraph, authors should add a statement that the present work is an observational study with no comparisons, correlations, and other statistical analyses regarding the fitness behavior of recruited subjects.

- Line 436: Since you state that age may be a factor that affects the generalization of the research findings to a greater size of population, you should provide data regarding the age of participants in this study as suggested in a previous comment.

- At the end of the discussion section, a separate paragraph titled “Conclusions” should be added underlining the main findings and suggesting future research attempts in this area (please move the statements from lines 437-439 to this paragraph).

Author Response

(The authors gave the same response as above.)

Reviewer 4 Report

Dear authors:

Although the document presents very interesting data, it looks more like a report, for the city of Beijing government, than a scientific paper. It has some methodological issues:  the objectives are not presented, the statistics in the results are not well explained, furthermore, the discussion is only a summary of the results, and finally, there are not conclusions.

Here there are some specific comments about the document.

Line 31. The abstract is not complete, it lacks objetives and conclusions.

Line 45: What do the authors mean by “fitness rate of residents aged 20-69 is only 14.7%”

Line 48: What do the authors mean by “rate of fitness”?

Line 53:  What do the authors mean by “REF”?

Line 57: What do the authors mean by urban fitness behavior?

Line 69:  Regarding Weibo, I have looked online and it explain that it is similar to Facebook and Tweeter but I believe authors should explain it in more detail in the article.

Line 92-93: How do the authors know this affirmation “It is the main place where residents' fitness behavior occurs.”?

Line 96-97: What is the exact number of subjects in this study? It said more than 13 million, but they should be more specific. Do the author have the participants’ consent to use their data for research purposes?

Line 163. I am not able to see the results in figure 1.

Regarding the results, the units are not presented in many of them. Furthermore, how do they do the statistics? in line 309 they say “significant hot-spots” How do the know it is significant? 

Kind regards,

Author Response

(The authors gave the same response as above.)

Round 2

Reviewer 4 Report

Dear authors,

Your article has better-quality, however it still, in my opinion, needs further improvements in order to be published, you have my new comments in blue in the attached document.

Kind regards,

Round 3

Reviewer 4 Report

Dear authors,

Congratulations on the improvement of your paper, here you have some details that can be improved.

Line 447 remain the reader what theme 3 and theme 4 are.

Lines 533 to 535. This sentence should be rewritten for a better understanding.

Third, residents tend to have a better experience when they perform fitness behaviors in the gym, while they tend to have a poor experience when they perform professional fitness in the gym. 

Kind regards,
